# Depletion of the Microbiota Has a Modest but Important Impact on the Fungal Burden of the Heart and Lungs during Early Systemic *Candida auris* Infection in Neutropenic Mice

**DOI:** 10.3390/microorganisms10020330

**Published:** 2022-02-01

**Authors:** Amber M. Pichowicz, Steven R. Torres, Fernando J. Torres-Velez, Adina D. Longyear, Navjot Singh, Erica Lasek-Nesselquist, Magdia De Jesus

**Affiliations:** 1College of Nursing and Health Sciences, University of Vermont, Burlington, VT 05405, USA; amberpichowicz97@gmail.com; 2Department of Biomedical Sciences, School of Public Health, University at Albany, Albany, NY 12208, USA; srtorress196@potsdam.edu (S.R.T.); adina.callejo@gmail.com (A.D.L.); 3Division of Infectious Diseases, Wadsworth Center, New York State Department of Health, Albany, NY 12208, USA; torres.fernandoj@gmail.com; 4Applied Genomics Core, Wadsworth Center, New York State Department of Health, Albany, NY 12208, USA; navjot.singh@health.ny.gov; 5Bioinformatics Core, Wadsworth Center, New York State Department of Health, Albany, NY 12208, USA; erica.lasek-nesselquist@health.ny.gov

**Keywords:** *Candida auris*, microbiome, germ-free, 1A8, neutropenia

## Abstract

The progression and systemic pathobiology of *C. auris* in the absence of a microbiota have not been described. Here, we describe the influence of the microbiota during the first 5 days of *C. auris* infection in germ-free or antibiotic-depleted mice. Depletion of the bacterial microbiota in both germ-free and antibiotic-depleted models results in a modest but important increase in the early stages of *C. auris* infection. Particularly the heart and lungs, followed by the cecum, uterus, and stomach, of intravenously (i.v.) infected neutropenic mice showed significant fungal organ burden. Understanding disease progression and pathobiology of *C. auris* in individuals with a depleted microbiota could potentially help in the development of care protocols that incorporate supplementation or restoration of the microbiota before invasive procedures, such as transplantation surgeries.

## 1. Introduction

For the past decade, the emerging fungal organism *Candida auris* has reminded us once more that similar to bacteria, fungi can become antimicrobial resistant. With a limited arsenal of antifungals and with few new antifungal candidates, surveillance becomes paramount [1,2]. Although *C. auris* was first isolated in 2009 from the inner ear canal of a Japanese patient, the first documented description was obtained from a South Korean retrospective study in 1996, where the organism was misidentified as *Candida haemulonii* [3,4]. At the time, species-level misidentification was due to the lack of *C. auris*-targeted methods achieved by PCR and MALDI-TOF [3,4,5,6]. For the past decade, infections caused by *C. auris* have been on the rise worldwide [3]. In 2016, the Centers for Disease Control and Prevention (CDC) released a clinical alert for healthcare facilities warning of the international emergence of *C. auris* infections with high mortality rates [3,7]. There are many hypotheses that speculate the reasons for *C. auris* emergence, such as global warming, the extensive use of broad-spectrum antimicrobials, animal reservoirs, increase in immunosuppressed patients, use of invasive devices, disrupted microbiomes, and underlying illness [8,9,10,11,12]. 

In recent years, the impact of the microbiome on the overall health both locally and systemically has also taken center stage [13,14]. It has been found that fungi are also an integral part of the microbiome and thus the term “mycobiome” has been used to describe these communities [15,16]. Recent modeling dynamics of the human mycobiome have provided insight into underlying alterations of the skin fungal community as a possible modifiable risk factor for acquisition and persistence of *C. auris* [17]. As of now, the systemic pathobiology of *C. auris* in the absence of a microbiota has not been described. Here, we describe the influence of the microbiota during the first 5 days of *C. auris* infection in germ-free or antibiotic-depleted mice. Additionally, we compared differences via the intravenous and gavage routes as well as the differences between immunocompetent and neutropenic mice treated with the monoclonal antibody (mAb 1A8). The results reveal that depletion of the bacterial microbiota in both germ-free and antibiotic-depleted models leads to a modest but important increase in the early stages of *C. auris* infection. Particularly the heart and lungs, followed by the cecum, uterus, and stomach, of intravenously (i.v.) infected neutropenic mice showed significant fungal organ burden. We also observed a modest increase of fungal burden in the liver of antibiotic-treated mice that had been infected via gavage. Collectively, these data suggest that the microbiota has a modest impact during early *C. auris* infection and that infection progression in this model could be further explored. 

## 2. Materials and Methods

### 2.1. Animal Use Ethics Statement

Animal research was reviewed and approved by the Wadsworth Center’s Institutional Animal Care and Use Committee (IACUC) under protocol #18-450. The Wadsworth Center complies with the Public Health Service Policy on Humane Care and Use of Laboratory Animals and was issued assurance number A3183-01. The Wadsworth Center is fully accredited by the Association for Assessment and Accreditation of Laboratory Animal Care (AAALAC). 

### 2.2. Animals and Microbiota Depletion

Germ-free BALB/c and BALB/c mice (6–8 weeks old) were obtained from Taconic Farms (Hudson, NY, USA). Germ-free mice were housed in sterile micro-isolator cages (Lab products, Seaford, DE, USA). The cages were maintained inside of a sterile biosafety cabinet for 5 days during the duration of the experiment; this was done as an alternative to a germ-free facility. There were *n* = 5 mice per group. To reduce the introduction of a microbiota, germ-free mice were provided a sterile gel pack in place of a water bottle and autoclaved, irradiated feed (Lab Diet, St. Louis, MO, USA). To deplete the microbiota, the mice were given a combination of 2 mg/mL streptomycin sulfate (Sigma-ALDRICH); 1500 U of penicillin G/mL (Sigma-Aldrich); and 50 mg/mL of gentamicin (Gibco) in their drinking water during the duration of the experiment. 

### 2.3. Neutrophil Depletion and Infection with C. auris

Neutrophil depletion was done as previously described [18]. Briefly, intraperitoneal (i.p.) injection of 200 μg of monoclonal antibody 1A8 that targets Ly6G cells (Bio X Cell, West Lebanon, NH, USA) was performed. The antibody injection was administered 24 h prior to infection and every 48 h thereafter. Tail vein and gavage infections were done with 10^7^ cells of *C. auris* strain CAU-09 acquired from the available Centers for Disease Control (CDC) panel. We chose to infect mice with the CAU-09 strain (S. Asian Clade I) for several reasons: We found it to be less aggregative than other *C. auris* strains, which provided ease of counting by a hemocytometer and prepare dilutions for tail vein infections. Clade I was the major genotype (>99%) among clinical isolates reported from New York, where the initial outbreaks were identified in the United States [19]. Lastly, we also used this strain to establish infection in a neutrophil-depleted model [18,20].

### 2.4. Fecal Pellet and Urine Collection

As previously described, fecal pellets were collected before initial neutrophil depletion, before infection, and prior to euthanasia. Then, 4–5 fecal pellets, approximately 180–220 mg, were collected for DNA isolation, snap-frozen in liquid nitrogen, and stored at 80 °C.

### 2.5. Tissue Collection for CFU Counts and Histology

As previously described, the mice were euthanized by CO_2_ and cardiac puncture [18]. Bladder, uterus, spleen, kidney, cecum, small intestine, stomach, liver, heart, lung, and brain necropsies were collected; weighed; and used for CFU counts and histology. For CFU counts, the organs were homogenized in Hanks Sodium Balanced Salt (HBSS) (Thermo Fisher, Waltham, MA, USA) through a 70 μm cell strainer (Cell Treat, Pepperell, MA, USA). A 1:100 dilution of organ homogenate was plated onto a Sabouraud dextrose agar plate with antibiotics (chloramphenicol (20 μg/mL), penicillin (20 U), gentamicin (40 μg/mL), and streptomycin (40 μg/mL)) [18].

### 2.6. Histopathology and Immunohistochemistry

As previously described, representative liver, kidney, heart, brain, stomach, spleen, lung, uterus, urinary bladder, cecum, and small and large intestine samples were collected during necropsy and fixed in 10% buffered formalin for 24 h [18]. The tissues were subsequently transferred to 70% ethanol prior to being processed and embedded. Sections were stained with hematoxylin and eosin for histopathological evaluation. For immunohistochemistry (IHC), 3–4 μm sections were deparaffinized in CitriSolve (Decon Labs., King of Prussia, PA, USA) and rehydrated by processing through graded alcohols. Tissues were pretreated with Proteinase K (10 μg/mL) for 15 min. Endogenous IgG and non-specific background were blocked with Rodent Block M (Biocare Medical; Pacheco, CA, USA) for 20 min, followed by an alkaline phosphatase block (BLOXALL; Vector Laboratories, Burlingame, CA, USA) for 10 min. The primary antibody (PA1-7206; Thermo Fisher Scientific, Waltham, MA, USA) was incubated on the tissue sections at a dilution of 1:10,000 for 1 h at room temperature. Sections were sequentially incubated with a rabbit-on-rodent tissue alkaline phosphatase-based polymer and Warp Red (Biocare Medical, Pacheco, CA, USA). Tissues were counterstained with Tacha’s hematoxylin and mounted using EcoMount (Biocare Medical, Pacheco, CA, USA) [18]. 

### 2.7. Fecal DNA Purification, PCR, and 16S rDNA Sequencing

DNA from the fecal samples was isolated using Qiagen AllPrep^®^ PowerFecal^®^ DNA/RNA Kit according to the manufacturer’s instructions. A modification made was that fecal samples were lysed with provided lysis buffer using a vertical Disruptor Genie for 5 min at 3000 rpm. Fecal DNA samples were measured by Qubit quantification. PCR reactions were performed using the 16S primer for bacterial genomes and the ITS2 primer for fungal genomes. Fecal DNA was also used for qPCR and RT-qPCR to detect C. auris presence and viability. A standard curve was produced by performing a serial dilution of 10^8^ to 10^1^ of the *C. auris* cells. These dilutions were then processed identically to the fecal samples collected to extract and purify DNA and RNA. A TaqMan qPCR reaction was run on the DNA samples using the ITS2 primer and probes for *C. auris*. The PerfeCTa Multiplex qPCR ToughMix was spiked with bicoid plasmid and respective primers after pipetting of the negative template control wells. Each sample and control were run in duplicate on a 7500 Fast instrument for 45 cycles. A standard curve was generated and used to calculate the relative concentrations of *C. auris* in the fecal pellets depending on their cycle threshold value. The fecal pellet DNA samples were run identically to the standard curve samples. 16S Ilumina sequencing was done as per the manufacturer’s instructions.

### 2.8. 16S rDNA Sequencing Analysis

BBMerge from BBMap v.37.93 (https://sourceforge.net/projects/bbmap/, accessed on 26 January 2022) were simultaneously quality-trimmed and merged paired-end reads under the following parameters: reads were required to be a minimum of 100 or 150 base pairs in length (16S and ITS2 sequences, respectively), with an average PHRED quality score of 20 and a quality cutoff of 20 per position. Merged reads were discarded if the overlap between them contained three times the expected number of errors. BBDuk from BBMap removed forward and reverse primers from the merged 16S and ITS2 sequences using a k-mer size of 15 and a minimum k-mer size of 11. Seqtk v.1.3 (https://github.com/lh3/seqtk, accessed on 26 January 2022) converted fastq files to a fasta format, and all sequences were merged into a single multi-fasta file for OTU assignment and taxonomic classification in NINJA-OPS v.1.5.1 [21]. Briefly, Bowtie2 v.2.1.0 [22] aligned 16S and ITS2 sequences to concatenated Greengenes [23] and ITS2 Unite databases (clustered at 97% identity) [24] using the parameters outlined by the NINJA-OPS manual (https://github.com/GabeAl/NINJA-OPS, accessed on 26 January 2022). NINJA-OPS then binned sequences into OTUs and assigned taxonomic information based on sequence alignments to the concatenated databases. Absolute abundances were summarized to the genus level in QIIME v.1.9.0 with the summarize_taxa.py script [25], and significant changes in taxon abundance were calculated in DESEq2 [26] in R v.3.5.2 (http://www.R-project.org, accessed on 26 January 2022).

### 2.9. Statistical Analysis

All data are expressed as the mean ± SEM. When comparing the means from two or more treatment groups, a nonparametric one-way analysis of variance (ANOVA) with a Kruskal–Wallis test, followed by an uncorrected Dunn’s multiple comparison test, was performed using Prism 6 (GraphPad La Jolla, CA). Statistical significance is indicated in figures using the following denotation: *p* > 0.05, * *p* ≤ 0.05, ** *p* ≤ 0.01, *** *p* ≤ 0.001, and **** *p* ≤ 0.0001.

## 3. Results

To determine the influence of the microbiota on *C. auris* infection, we compared the presence of *C. auris* in wild-type to germ-free or antibiotic-depleted mice. We measured fungal organ burdens during early infection (5 days post infection) in mice that were infected with an inoculum of 10^7^
*C. auris* cells. For comparison, we also infected mice via the intravenous and gavage routes. Since we had previously described the impact of *C. auris* infection on a neutropenic murine model using the anti-Ly6G^+^ monoclonal antibody 1A8 and determined that neutropenia is a good model to study infection progression, here we also used the neutropenic model to address how the microbiota influences early infection. Liver, heart, lung, kidney, bladder, uterus, spleen, cecum, small intestine, stomach, and brain necropsies were collected for assessment of fungal organ burden and histological analysis. Analysis of fungal organ burdens revealed that for the spleen, kidneys, the bladder, the small intestine, and the brain, the depletion of the microbiota did not have a statistically significant impact on mice, regardless of the infection route or the immune status. However, in the heart, lungs, the cecum, the uterus, and the stomach, all infected via the i.v. route, there were some notable but modest differences. In the liver, we found modest differences in mice that had been gavaged with *C. auris*. In the heart, a threefold difference was observed in fungal organ burdens for neutropenic germ-free mice at 10^6^ CFU/g in comparison to 10^3^ CFU/g in germ-free wt. control mice. In the lungs, we found a twofold change. In the cecum, stomach, and uterus of i.v.-infected, antibiotic-treated mice, we found a modest threefold change. In the liver, we found a twofold change only in gavaged, neutropenic mice (Figure 1). ITS2 sequencing of i.v. and gavage, germ-free models revealed that regardless of the infection route, the dominant fungal species in feces was *C. auris* (Figure 2).

To further access the pathobiology of *C. auris* in the harvested organs, we evaluated and scored *C. auris*-infected tissues. The results revealed that the organs with the highest immune activity were the heart, kidneys, and liver of i.v.-infected germ-free mice as these contained prominent extramedullary hematopoiesis, intracytoplasmic staining of numerous tissue macrophages, abscesses, and yeast cells. The lungs and the spleen also contained some tissue macrophages and yeast cells but not to the same degree as the aforementioned organs (Table 1). In mice with antibiotic-depleted microbiota, the organs with the most severe multifocal to coalescing inflammation, infiltration of macrophages with necrotizing microabscessation, and abscessation were the heart and kidneys, with some tissue macrophages found in the lungs, the spleen, and the liver. In both germ-free and microbiota-depleted models, we found that the gavage route of infection had little impact in terms of establishing infection that leads to pathogenesis, suggesting the rapid clearance of *C. auris* through feces (Table 2 and Figure 2). 

## 4. Discussion

Although *C. auris* can colonize humans, the risk factors are similar those of other types of *Candida* infections and include individuals who have compromised immunity, such as those with cancer, diabetes, transplants, or a history of abdominal surgery; those having central venous catheters; and those on ventilators or prolonged antibiotic treatment [10,27]. Case studies have highlighted that patients who succumbed to systemic *C. auris* infections often exhibited heart failure, renal failure, respiratory arrest, pneumonia, and multiorgan failure [2,28,29,30]. Because the disruption of the microbiome has been associated with inflammation or disease, we tested the role of the microbiota during early *C. auris*. 

Some limitations of this study were that we only looked at early *C. auris* infection, within the first five days, and that we used the CAU-09 strain for infections. Early infection was chosen to capture the sequential progression of fungal organ burden. Additionally, since our germ-free mice were inside of a biosafety cabinet, we could not extend the duration of the experiment without the risk of introducing external microbes. We chose to use the *C. auris* CAU-09 S. Asian strain for several reasons: It frequently causes outbreaks of invasive infection, such as those documented in hospital settings across New York State, United States, since 2013 [19,31]. We also found CAU-09 to be less aggregative than other *C. auris* strains, which provided counting accuracy by hemocytometer and helped prepare dilution suspensions for tail vein infections [32]. Lastly, we had previously developed a neutropenic model that was tested with CAU-09, which would provide a baseline to compare when we depleted the microbiota [18]. For this model, we found that the heart, lungs, cecum, uterus, and stomach of intravenously (i.v.) infected neutropenic mice showed significant fungal organ burden. Susceptibility of the heart and lungs in a microbiota-depleted state is particularly interesting because it provides information that can be used in healthcare facilities. For example, a recent study suggests that candidates for heart transplantations should require early detection and treatment for *C. auris* as the open wounds required for heart surgery could lead to systemic infection [33]. The study went on to highlight that *C. auris* colonization in heart transplant patients may be a contraindication [33]. In another study, donor-derived transmission of *C. auris* during lung transplantation was documented in an immunosuppressed patient receiving antibiotic and antifungal therapy [2]. Understanding that a depleted microbiota also increases the risk for *C. auris* infections can help in the better development of care protocols that can incorporate supplementation or restoration of the microbiota before transplantation surgery in these high-risk patients. 

## 5. Conclusions

In this study, we observed that the absence of a microbiota in germ-free or antibiotic-depleted murine models has a modest impact on early *C. auris* infection, by the assessment of fungal organ burden. We compared differences via the intravenous and gavage routes as well as the differences between immunocompetent and neutropenic mice treated with the monoclonal antibody (mAb 1A8). The heart and lungs, followed by the cecum, the uterus, and the stomach, of intravenously (i.v.) infected neutropenic mice showed significant fungal organ burden. We also observed a modest increase in the liver of antibiotic-treated mice that had been infected via gavage. These data suggest that the microbiota has a modest but important impact during early *C. auris* infection. Infection progression highlighted in this model could be further followed in high-risk immunocompromised individuals on extensive antibiotic therapies who may require organ transplantation. 

## Figures and Tables

**Figure 1 microorganisms-10-00330-f001:**
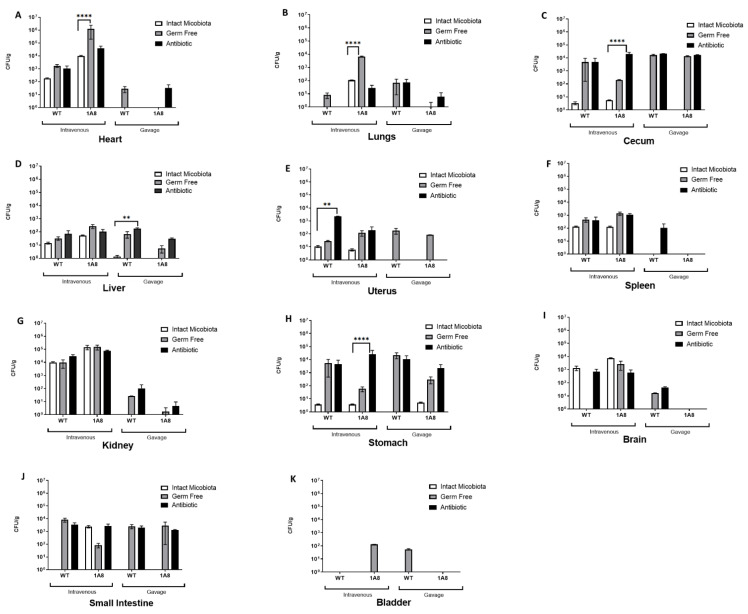
Depletion of the microbiota has minimal impact on *C. auris* infection. (**A**–**K**) Fungal organ burden in germ-free and microbiota-depleted mice infected by i.v. and gavage with 10^7^ cells of *C. auris* routes. Organ fungal burden is measured in CFU/g in tissues, 5 days post infection, *n* = 5 mice per group. A nonparametric one-way analysis of variance (ANOVA) with a Kruskal–Wallis test, followed by an uncorrected Dunn’s multiple comparison test, was carried out using ns *p* > 0.05, ** *p* ≤ 0.01, and **** *p* ≤ 0.0001.

**Figure 2 microorganisms-10-00330-f002:**
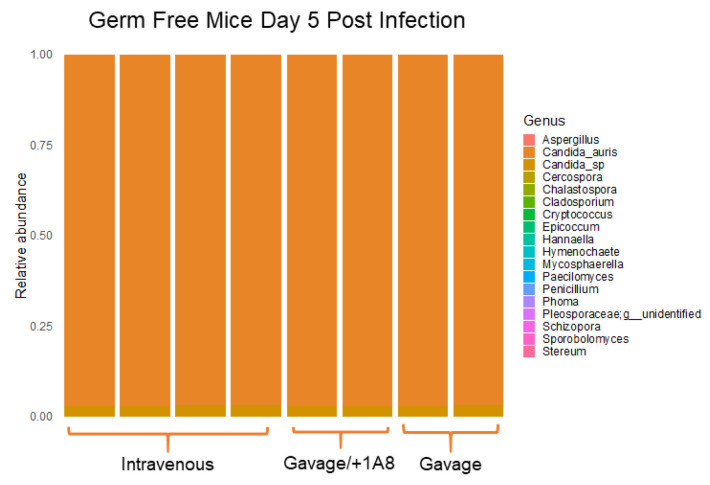
The dominant species in the fecal microbiota is *C. auris* in germ-free mice at 5 days post infection. Fecal DNA was extracted and 16S rDNA and ITS2 libraries were sequenced via Illumina technology. ITS2 sequencing in fecal microbiota demonstrates that *C. auris* is the dominant species.

**Table 1 microorganisms-10-00330-t001:** Fungal organ burden scoring system for neutrophil germ-free mice infected with *C. auris*. A comparison of the routes of infection and neutrophil depletion shows the following: Lesion scoring (HE): - within normal limits (WNL); + minimal to mild suppurative inflammation, scattered macrophages, and rare microabscessation; ++ mild to moderate suppurative inflammation, moderate numbers of macrophages, and microabscesses; +++ severe multifocal necrotizing suppurative inflammation, numerous macrophages, and microabscesses; ++++ severe multifocal to coalescing necrotizing suppurative inflammation, numerous macrophages, and microabscesses.

Sample	Liver		Kidney		Spleen		Heart		Uterus		Lung		Urinary Bl		Brain		Stomach		Small Intestine		Large Intestine	
	*HE*	*IHC*	*HE*	*IHC*	*HE*	*IHC*	*HE*	*IHC*	*HE*	*IHC*	*HE*	*IHC*	*HE*	*IHC*	*HE*	*IHC*	*HE*	*IHC*	*HE*	*IHC*	*HE*	*IHC*
Control	-	-	-	-	-	-	-	-	-	-	-	-	-	-	-	-	-	-	NA	NA	NA	NA
Control	-	-	-	-	-	-	-	-	-	-	-	-	-	-	-	-	-	-	NA	NA	NA	NA
Control	-	-	-	-	-	-	-	-	-	-	-	-	-	-	-	-	-	-	-	-	-	-
i.v.	++	++	+	+	-	++	++	++	-	-	-	-	NA	NA	+	+	-	+	NA	NA	NA	NA
i.v.	+	++	-	+	-	+	+	++	-	-	-	-	-	-	-	-	-	+	NA	NA	NA	NA
i.v.	+	++	+	+	-	+	+	+	-	-	-	-	-	-	-	+	-	+	-	+	-	+
i.v + 1A8	+	+++	+++	+++	-	++	++++	++++	-	+	-	++	+	+	+	+	+	+	NA	NA	NA	NA
i.v + 1A8	+	+++	+++	+++	-	++	++++	++++	-	-	-	+	-	-	-	+	+	+	NA	NA	NA	NA
i.v + 1A8	+	+++	+++	+++	-	++	++++	++++	-	^§^	-	+	-	-	-	+	-	+	++	+	-	+
Gavage	-	-	-	-	-	-	-	-	-	-	-	-	-	-	-	-	-	+	NA	NA	NA	NA
Gavage	-	-	-	-	-	-	-	-	-	-	-	+	-	-	-	-	-	+	NA	NA	NA	NA
Gavage	-	-	-	-	-	-	-	-	-	-	-	-	NA	NA	-	-	-	+	-	+	-	+
Gavage + 1A8	-	-	-	-	-	-	-	-	-	-	-	-	-	-	-	-	-	-	NA	NA	NA	NA
Gavage + 1A8	EMH	-	-	-	-	-	-	-	-	-	-	++	-	-	-	-	-	-	NA	NA	NA	NA
Gavage + 1A8	-	-	-	-	-	-	-	-	-	-	-	-	-	-	-	-	-	-	-	+	-	+

Severity of histopathological changes as observed in hematoxylin and eosin (H&E) stain: (-) within normal limits; (+) mild extramedullary hematopoiesis (EMH) with a distinct expansion of the megakaryoblast and myeloblast lines (liver) with rare macrophage infiltration and microgranulomas (liver); (++) prominent EMH of the megakaryoblast and myeloblast lines (liver), macrophage infiltration admixed with few necrotic cells, and rare microgranulomas (liver); (+++) extensive macrophage infiltration admixed with few necrotic cells; (++++) prominent macrophage infiltration and necrosis with numerous intralesional fungal elements. Extent and distribution of *C. auris antigen* (immunohistochemistry): (-) no antigen detected; (+) intracytoplasmic staining of a few tissue macrophages, granulomas, meninges, and rare yeast; (++) intracytoplasmic staining of numerous tissue macrophages and tubular epithelial cells, granulomas, meninges, and yeasts; (+++) intracytoplasmic staining of numerous tissue macrophages, granulomas, tubular epithelial cells, and meninges. Prominent staining of numerous intralesional yeasts and necrotic debris. ^§^ Positive granulomas in peritoneal fat.

**Table 2 microorganisms-10-00330-t002:** Fungal organ burden scoring system for neutrophil microbiota-depleted mice infected with *C. auris*. A comparison of the routes of infection and neutrophil depletion shows the following: Lesion scoring (HE): - within normal limits (WNL); + minimal to mild suppurative inflammation, scattered macrophages, and rare microabscessation; ++ mild to moderate suppurative inflammation, moderate numbers of macrophages, and micro abscesses; +++ severe multifocal necrotizing suppurative inflammation, numerous macrophages, and microabscesses; ++++ severe multifocal to coalescing necrotizing suppurative inflammation, numerous macrophages, and microabscesses.

Sample	Liver		Kidney		Heart		Brain		Stomach		Spleen		Lung		Uterus		Urinary Bladder		Small Intestine		Cecum/Large Intestine	
	*HE*	*IHC*	*HE*	*IHC*	*HE*	*IHC*	*HE*	*IHC*	*HE*	*IHC*	*HE*	*IHC*	*HE*	*IHC*	*HE*	*IHC*	*HE*	*IHC*	*HE*	*IHC*	*HE*	*IHC*
Control	-	-	-	-	-	-	-	-	-	-	-	-	-	-	NT	NT	NT	NT	NT	NT	NT	NT
Control	-	-	-	-	-	-	-	-	-	-	-	-	-	-	-	-	NT	NT	NT	NT	NT	NT
Control	-	-	-	-	NT	NT	-	-	-	-	NT	NT	-	-	NT	NT	NT	NT	-	-	-	-
i.v.	+	-	++	+	+	+	-	-	-	-	-	-	-	-	NT	NT	-	+	NT	NT	NT	NT
i.v.	-	-	++	+	+	+	-	-	-	-	-	+	Au	-	-	-	-	-	NT	NT	NT	NT
i.v.	-	-	NT	NT	+	++	-	-	-	-	-	-	Au	-	NT	NT	NT	NT	-	-	-	+
i.v + 1A8	+	+	+	++	+++	+++	-	+	-	-	-	+	-	+	-	-	-	-	NT	NT	NT	NT
i.v + 1A8	+	+	+++	+++	+++	+++	-	-	-	+	-	+	-	-	NT	NT	-	-	NT	NT	NT	NT
i.v + 1A8	+	+	+++	+++	+++	+++	+	++	-	-	-	+	-	+	-	-	-	-	-	-	-	+
Gavage	-	-	-	-	-	-	-	-	-	-	-	-	-	-	-	-	-	-	NT	NT	NT	NT
Gavage	-	-	-	-	-	-	-	-	-	-	-	-	-	-	-	-	-	-	NT	NT	NT	NT
Gavage	-	-	-	-	-	-	-	-	-	-	-	-	-	-	-	-	-	-	-	+	-	+
Gavage + 1A8	-	-	-	-	-	-	-	-	-	-	-	-	-	-	-	-	NT	NT	NT	NT	NT	NT
Gavage + 1A8	-	-	-	-	-	-	-	-	-	-	-	-	-	-	-	-	-	-	NT	NT	NT	NT
Gavage + 1A8	-	-	-	-	-	-	-	-	-	-	-	-	-	-	-	-	-	-	-	+	-	+

Lesion scoring (HE): - WNL; + minimal to mild inflammation, infiltration of neutrophils, and rare macrophages with rare microabscessation; ++ mild to moderate inflammation, infiltration of neutrophils and macrophages, apoptosis or inflammatory and parenchymal cells, and microabscessation, often with a necrotic core; +++ severe multifocal necrotizing inflammation, infiltration of neutrophils and macrophages with necrotizing microabscessation, and abscessation; ++++ severe multifocal to coalescing inflammation, infiltration of neutrophils and macrophages with necrotizing microabscessation, and abscessation. IHC: - negative; + 0-5 cells or yeasts per HPF (50x), macrophages, and rare yeast; ++ greater than 5 cell/HPF, intralesional in abscesses, yeast, and scattered antigen in the debris and intracytoplasmic in macrophages; +++ numerous cells/HPF, intralesional in abscesses, numerous yeasts, and scattered antigen in the debris and intracytoplasmic in macrophages; ++++ numerous cells/HPF, intralesional in abscesses, copious amounts of yeast, and scattered antigen in the debris and intracytoplasmic in macrophages. NT: Tissue not available for evaluation. Au: Marked autolysis.

## Data Availability

Not Applicable.

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
