# Peer review of "Depletion of the Microbiota Has a Modest but Important Impact on the Fungal Burden of the Heart and Lungs during Early Systemic Candida auris Infection in Neutropenic Mice"

_microorganisms, 2022, doi:10.3390/microorganisms10020330_

Round 1

Reviewer 1 Report

The manuscript #microorganisms-1555668, entitled “Depletion of the microbiota has a modest impact in heart and lung fungal burden during early systemic Candida auris infecttion in neutropenic mice” by Pichowicz et al. presents inoculation of germ-free mice with C. auris and further investigation of its spread to inner organs throughout the 5-day period. The authors conclude that the absence of microbiota has a modest impact on the spread of the infection. The topic is of a great importance as Candida spp. infections still display high mortality among immunocompromised patients, as well as drug resistance towards most commonly used azoles. The latter trait is clearly seen in C. auris as this species is known to possess natural resistance towards the azoles and other antifungal drugs. Thus, the manuscript brings scientific novelty and touches an important issue. The authors were quite critical in discussing/conluding their own study. In my opinion, however, the authors must discuss the fact that they were using only one C. auris strain and the acquired results might have been quite different for another strain. Intra-strain alternations within basic metabolism/physiology/virulence etc. among strains of the same Candida sp. is being constantly reported. 

Additionally, the edition/redaction of the ms must be performed. Some notable examples are listed below:

The title in the susy contains "infecttion";

numerous examples of double spaces are present;

include space between the values and the units;

in some parts C. auris is not italicized;

line 14-15: "in both in a germ";

line 79 and 156, the unit must be included next to "107";

Figure 1 is illegible; the plots should have been bigger, whereas the letters "A,B,C" smaller

Author Response

The title in the susy contains "infecttion"; Response: It was correct in manuscript but could not correct in journal entry system.

numerous examples of double spaces are present;

include space between the values and the units; Response: Spaces adjusted

in some parts C. auris is not italicized; Response: C.auris throughout manuscript italicized

line 14-15: "in both in a germ"; Response: Corrected

line 79 and 156, the unit must be included next to "107"; Response: Corrected to include cells as the unit.

Figure 1 is illegible; the plots should have been bigger, whereas the letters "A,B,C" smaller Response: Made the graphic slightly larger and letters smaller.

Reviewer 2 Report

The authors study the role of the microbiota in the control of C auris infection. In particular, the results describe the impact of microbiota during the first 5 days of C. auris infection.                                                                 The issue is interesting for the relevance of the C. auris infection in human pathology but it is appropriate to provide same further information on:         the virulence characteristics of strain used for the experiments before and after infection;                                                                                                   the number of animals used for each experimental condition to  demonstrate reproducibility.                                                                         The authors have to modify trough the manuscript C. auris in Italic  C.auris. 

The conclusion has to be improved  adding the reasons for the choice of C. auris strain for the study rather than others Candida spp.

and the references including the following papers:

Teresa Fasciana, Andrea Cortegiani, Mariachiara Ippolito, Antonino Giarratano, Orazia Di Quattro, Dario Lipari,Domenico Graceffa and Anna Giammanco Candida auris: An Overview of How to Screen, Detect, Test and Control This Emerging Pathogen
Antibiotics 2020, 9(11), 778; https://doi.org/10.3390/antibiotics9110778

Pandya, N., Cag, Y., Pandak, N., ...Khan, E.A., Erdem, H. International multicentre study of Candida auris infections Journal of Fungi, 2021, 7(10), 878

Author Response

The conclusion has to be improved  adding the reasons for the choice of C. auris strain for the study rather than others Candida spp.

Response: We have added the reasons for the choice of C.auris strain in the dicussion. It now reads
“We chose to use the C. auris CAU-09 S. Asian strain for several reasons, as it frequently causes outbreaks of invasive infection such as those documented in hospital settings across New York State, United States since 2013 [19,31]. We also found CAU-09 to be less aggregative than other C. auris strains which provided counting accuracy by he-mocytometer,and prepare dilution suspensions for tail vein infections [32]. Lastly, we had previously developed a neutropenic model that was tested with CAU-09, this would provide a baseline to compare when we depleted the microbiota[18]. For this model, we found that the heart, lung, cecum, uterus and stomach of intravenously (i.v) infected neutropenic mice showed significant fungal organ burden.”

and the references including the following papers:

Teresa Fasciana, Andrea Cortegiani, Mariachiara Ippolito, Antonino Giarratano, Orazia Di Quattro, Dario Lipari,Domenico Graceffa and Anna Giammanco Candida auris: An Overview of How to Screen, Detect, Test and Control This Emerging Pathogen
Antibiotics 2020, 9(11), 778; https://doi.org/10.3390/antibiotics9110778

Pandya, N., Cag, Y., Pandak, N., ...Khan, E.A., Erdem, H. International multicentre study of Candida auris infections Journal of Fungi, 2021, 7(10), 878

Response: The suggested references were added to the manuscript.